# Machine Learning Methods to Identify Predictors of Psychological Distress

**Yang Chen** [1], **Xiaomei Zhang** [1], **Lin Lu** [2], **Yinzhi Wang** [1], **Jiajia Liu** [3], **Lei Qin** [1], **Linglong Ye** [4], **Jianping Zhu** [5,6], **Ben-Chang Shia** [7,8,*]  **and Ming-Chih Chen** [7,8,*] 

[1] School of Statistics, University of International Business and Economics, Beijing 100029, China; 17862922732@163.com (Y.C.); zhangxmuibe@163.com (X.Z.); 202011422191@uibe.edu.cn (Y.W.); qinlei@uibe.edu.cn (L.Q.)
[2] Institute of Education and Economy Research, University of International Business and Economics, Beijing 100029, China; lulin_2013@163.com
[3] School of International Relations, University of International Business and Economics, Beijing 100029, China; 13930777788@139.com
[4] School of Public Affairs, Xiamen University, Xiamen 361005, China; leyloria@gmail.com
[5] School of Management, Xiamen University, Xiamen 361005, China; xmjpzhu@163.com
[6] National Institute for Data Science in Health and Medicine, Xiamen University, Xiamen 361005, China
[7] Graduate Institute of Business Administration, College of Management, Fu Jen Catholic University, New Taipei City 24205, Taiwan
[8] Artificial Intelligence Development Center, Fu Jen Catholic University, New Taipei City 24205, Taiwan
* Correspondence: 025674@mail.fju.edu.tw (B.-C.S.); 081438@mail.fju.edu.tw (M.-C.C.)

**Abstract:** As people pay ever-increasing attention to the problems caused by psychological stress, research on its influencing factors becomes crucial. This study analyzed the Health Information National Trends Survey (HINTS, Cycle 3 and Cycle 4) data (N = 5484) and assessed the outcomes using descriptive statistics, Chi-squared tests, and *t*-tests. Four machine learning algorithms were applied for modeling: logistic regression (linear), random forests (RF) (ensemble), the artificial neural network (ANN) (nonlinear), and gradient boosting (GB) (ensemble). The samples were randomly assigned to a 50% training set and a 50% validation set. Twenty-six preselected variables from the databases were used in the study as predictors, and the four models identified twenty predictors of psychological distress. The essence of this paper is a binary classification problem of judging whether an individual has psychological distress based on many different factors. Therefore, accuracy, precision, recall, F1-score, and AUC were used to evaluate the model performance. The logistic regression model selected predictors by forward selection, backward selection, and stepwise regression; variable importance values were used to identify predictors in the other three machine learning methods. Of the four machine learning models, the ANN exhibited the best predictive effect (AUC = 73.90%). A range of predictors of psychological distress was identified by combining the four machine learning models, which would help improve the performance of the existing mental health screening tools.

**Keywords:** psychological distress; predictors; machine learning; HINTS

## 1. Introduction

Psychological distress is a state of emotional suffering associated with stressors and demands that are difficult to cope with in daily life [1], and describes an acute stress disorder caused by a living environment or a mental health disorder. Surveys have shown that psychological distress may lead to emotional instability and interpersonal difficulties, and severe psychological distress can disrupt the body's biological rhythm, even causing fatal diseases. However, the difficulty in identifying psychological distress is frustrating for patients and health professionals alike. At present, psychological tests or hormone tests are

carried out to detect psychological distress, but potential patients with psychological distress seldom take the initiative to undergo any professional testing. Therefore, identifying predictors and reaching a timely diagnosis is beneficial to public mental health.

Considering the current research on the factors influencing psychological distress, some research applied traditional statistical methods to explore the relationship between certain factors and psychological distress. Weaver et al. (1995) examined the relationship between interpersonal violence and psychological distress through the descriptive statistics and statistical tests of the questionnaire data [2]. Kessler et al. (1998) used survival models to investigate the probability and time association between psychological distress and marital status [3]. Zabora et al. (2001) determined the prevalence of psychological distress in cancer patients, where univariate and multiple regression analyses were used to examine the relationship between relevant variables and psychological distress [4]. Additionally, Mirowsky et al. (2017) explored the impact of social stratification on psychological distress [5]. Drapeau et al. (2012) critically reviewed the empirical evidence on risk and protective factors associated with psychological distress in the general population and in two specific populations by constructing a scale [6]. Winefield et al. (2017) explored a self-report measure for psychological well-being and used factor analysis to investigate the relationship between mental health and psychological distress [7]. These studies mainly explored the influence of a certain factor or a class of factors on psychological distress, or only involved certain groups of people; therefore, the scope of the research was relatively limited.

With the rapid development of artificial intelligence, machine learning methods have received increasing attention. Machine learning algorithms are used in a wide variety of applications, such as in medicine and healthcare, where it is difficult or unfeasible to develop conventional algorithms to necessary tasks [8]. The second important role of machine learning in healthcare is to increase diagnostic accuracy, as machine learning can provide excellent capabilities to predict diseases [9]. For example, De Silva et al. (2020) used machine learning to identify predictors of prediabetes in a nationally representative sample of the U.S. population. The results demonstrated the value of machine learning in identifying a wide range of predictors that could enhance prediabetes prediction and clinical decision-making [10]. There are also articles on the use of machine learning methods to study psychological distress. Zhou X et al. (2006) used artificial neural networks and machine learning models to predict the incidence of psychological distress in Alzheimer's patients and achieved a relatively high prediction accuracy rate [11]. In Prout TA et al. (2020), a random forest machine learning algorithm was used to identify the strongest predictors of psychological distress during COVID-19, and regression trees were developed to identify individuals at greater risk for anxiety, depression, and post-traumatic stress. The random forest method is able to identify the most important predictors from a large set of potential predictor variables. Moreover, the subsequent regression tree analysis allows for the identification of various interactions between the predictor variables [12]. Sutter B et al. (2021) aimed to provide a foundation by building a machine learning model across multiple techniques to predict psychological distress from ecological factors alone, and eight different classification techniques were implemented on a sample dataset [13]. Using machine learning algorithms is likely to enhance a timely diagnosis of psychological distress. However, in these machine learning method studies on psychological distress, the data used were relatively limited, such as only data for certain disease groups or a certain period of time.

In this paper, we used data from the Health Information National Trends Survey (HINTS). The Health Information National Trends Survey (HINTS) is a probability-based and nationally representative survey of the U.S. adult (age 18+) noninstitutionalized population conducted by the NCI. HINTS regularly collects nationally representative data about the American public's knowledge of, attitudes toward, and use of cancer- and health-related information and provides a rich multidimensional data source for predictive analytics. Moreover, we applied some machine learning algorithms to a nationally representative

sample to optimize psychological distress prediction. According to our best knowledge, this is the first study that applied a range of machine learning algorithms to such a large representative sample based on many different factors (predictors of psychological distress). We implemented a combination of machine learning methods and authoritative data. Predictors of psychological distress can be identified based on the results of machine learning methods.

## 2. Data Source

Data for this study were collected from the National Cancer Institute's 2019–2020 Health Information National Trends Survey (HINTS). HINTS is a probability-based and nationally representative survey of the U.S. adult (age 18+) noninstitutionalized population conducted by the NCI. The purpose of creating a population survey is to track trends in the public's rapidly changing use of new communication technologies while charting progress in meeting health communication goals in terms of the public's knowledge, attitudes, and behaviors [14]. This study analyzed merged data from Cycle 3 to Cycle 4. Data from Cycle 3 were collected between January and May 2019, and data from Cycle 4 were collected from February 2019 to June 2019. We screened the respondents based on the target-dependent variable (i.e., Psychological Distress) and 26 potential independent variables (Gender, Age, Race, etc.; presented in Table 1), leaving the respondents with no missing values in all the variables. Finally, 5484 respondents were screened out, including 2956 and 2528 individuals in 2019 and 2020, respectively. Administration of HINTS was approved by the Institutional Review Board at Westat Inc. and deemed exempt by the National Institutes of Health Office of Human Subjects Research. Additional information on the survey design is available on the HINTS website, including weighting to allow respective national estimates and obtain accurate standard errors for statistical testing.

**Table 1.** Distribution of the characteristics of the 26 extracted variables in the HINTS database.

| Variable<br><br>Categorical Variables | Individuals without<br>Psychological Distress<br>n (%) | Individuals with<br>Psychological Distress<br>n (%) | *p*-Value |
|---|---|---|---|
| Total | 2610 (47.59) | 2874 (52.41) | |
| Gender | | | |
| Male | 1255 (48.08) | 1101 (38.31) | <0.0001 |
| Female | 1355 (51.92) | 1773 (61.69) | |
| Race | | | |
| Non-Hispanic white | 1687 (64.64) | 1901 (66.14) | 0.2408 |
| Racial and ethnic minority | 923 (35.36) | 973 (33.86) | |
| Education | | | |
| >High school | 2119 (81.19) | 2328 (81.00) | 0.8608 |
| ≤High school | 491 (18.81) | 546 (19.00) | |
| Income | | | |
| ≥$20,000 | 2333 (89.39) | 2392 (83.23) | <0.0001 |
| <$20,000 | 277 (10.61) | 482 (16.77) | |
| Area | | | |
| Metropolitan | 2324 (89.04) | 2572 (89.49) | 0.5908 |
| Non-metropolitan | 286 (10.96) | 302 (10.51) | |
| Marital status | | | |
| In marriage | 1647 (63.10) | 1550 (53.93) | <0.0001 |
| Not in marriage | 963 (36.90) | 1324 (46.07) | |
| SeekCancerInfo | | | |
| Yes | 1305 (50) | 1675 (58.28) | <0.0001 |
| No | 1305 (50) | 1199 (41.72) | |
| UseInternet | | | |
| Yes | 2280 (87.36) | 2607 (90.71) | <0.0001 |
| No | 330 (12.64) | 267 (9.29) | |

Table 1. *Cont.*

| Variable<br>Categorical Variables | Individuals without<br>Psychological Distress<br>n (%) | Individuals with<br>Psychological Distress<br>n (%) | *p*-Value |
|---|---|---|---|
| WearableDevTrackHealth | | | |
| Yes | 761 (29.16) | 886 (30.83) | 0.1777 |
| No | 1849 (70.84) | 1988 (69.17) | |
| Social media user | | | |
| Yes | 1935 (74.14) | 2410 (83.86) | <0.0001 |
| No | 675 (25.86) | 464 (16.14) | |
| RegularProvider | | | |
| Yes | 1843 (70.61) | 2018 (70.22) | 0.7475 |
| No | 767 (29.39) | 856 (29.78) | |
| AccessOnlineRecord | | | |
| Yes | 1152 (44.14) | 1349 (46.94) | 0.0376 |
| No | 1458 (55.86) | 1525 (53.06) | |
| Caregiving | | | |
| Yes | 367 (14.06) | 523 (18.20) | <0.0001 |
| No | 2243 (85.94) | 2351 (81.80) | |
| GeneralHealth | | | |
| Good | 2442 (93.56) | 2338 (81.35) | <0.0001 |
| Poor | 168 (6.44) | 536 (18.65) | |
| OwnAbilityTakeCareHealth | | | |
| Completely confident | 825 (31.61) | 487 (16.95) | <0.0001 |
| Very confident | 1376 (52.72) | 1332 (46.35) | |
| Somewhat confident | 386 (14.79) | 864 (30.06) | |
| A little confident | 19 (0.73) | 145 (5.05) | |
| Not confident at all | 4 (0.15) | 46 (1.60) | |
| Deaf | | | |
| Yes | 140 (5.36) | 211 (7.34) | 0.0028 |
| No | 2470 (94.64) | 2663 (92.66) | |
| TalkHealthFriends | | | |
| Yes | 2116 (81.07) | 2383 (82.92) | 0.0758 |
| No | 494 (18.93) | 491 (17.08) | |
| MedConditions_Disease | | | |
| Yes | 1401 (53.68) | 1992 (69.31) | <0.0001 |
| No | 1209 (46.32) | 882 (30.69) | |
| Drink | | | |
| Yes | 1320 (50.57) | 1501 (52.23) | 0.2215 |
| No | 1290 (49.43) | 1373 (47.77) | |
| Smoke | | | |
| Yes | 932 (35.71) | 1283 (44.64) | <0.0001 |
| No | 1678 (64.29) | 1591 (55.36) | |
| EverHadCancer | | | |
| Yes | 199 (7.62) | 180 (6.26) | 0.0472 |
| No | 2411 (92.38) | 2694 (93.74) | |
| Cancercheck | | | |
| Yes | 2604 (99.77) | 2864 (99.65) | 0.4182 |
| No | 6 (0.23) | 10 (0.35) | |
| FreqWorryCancer | | | |
| Not at all | 647 (24.79) | 371 (12.91) | <0.0001 |
| Slightly | 822 (31.49) | 819 (28.50) | |
| Somewhat | 699 (26.78) | 902 (31.38) | |
| Moderately | 313 (11.99) | 535 (18.62) | |
| Extremely | 129 (4.94) | 247 (8.59) | |
| Numeric variables | Mean (SD) | Mean (SD) | *p*-value |
| Age | 56.46 (0.31) | 50.65 (0.31) | <0.0001 |
| BMI | 28.06 (0.12) | 29.02 (0.13) | <0.0001 |
| ModerateExerciseMinutes | 45.07 (1.10) | 38.08 (0.96) | <0.0001 |

Note: Chi-squared tests for categorical variables and two-tailed *t*-test for continuous variables; level of significance: $p = 0.05$; $p < 0.05$ indicates a significant difference in this variable between individuals with and without psychological distress.

## 3. Statistical Analysis

This study compared the sociodemographic characteristics and related variables in individuals with or without psychological distress via Chi-squared tests for categorical variables and two-tailed *t*-tests for continuous variables. Four machine learning algorithms were applied for modeling: logistic regression (linear), random forests (RF) (ensemble), the

artificial neural network (ANN) (nonlinear), and gradient boosting (GB) (ensemble). To evaluate the predictive accuracy of the models, we randomly assigned 50% of the dataset to a training set and the remaining 50% to the validation set, reporting the accuracy, precision, recall, F1-score, and AUC of the validation set. The logistic regression model selected predictors by forward selection, backward selection, and stepwise regression. The relative effects of the predictors in the logistic regression model were measured by adjusted odds ratios (ORs), while the variability and significance were assessed by confidence intervals (CIs) and the corresponding *p*-values. Variable importance values were used in the other three classification algorithms to identify the predictors.

All statistical analyses were performed on R Software version 4.1.2. R is a programming language for statistical computing and graphics created by statisticians Ross Ihaka and Robert Gentleman.The official R software environment is an open-source free software environment within the GNU package, available under the GNU General Public License. The $p < 0.05$ was considered statistically significant.

## 4. Measures

### 4.1. Psychological Distress

Psychological distress is an emotional state associated with intractable stressors and demands in daily life, with depression and anxiety as its manifestations. The variable "Psychological Distress" in this paper was calculated by the HINTS using the following four items. The first two items are for depression screening, with the other two for anxiety screening: Over the past two weeks, how often have you been bothered by any of the following problems? (a) Little interest or pleasure in doing things, (b) Feeling down, depressed, or hopeless, (c) Feeling nervous, anxious, or on edge, and (d) Not being able to stop or control worrying. There were four answer choices for cases (a) to (d): (1) Nearly every day, (2) More than half the days, (3) Several days, and (4) Not at all. We reclassified the answers into two categories, whereby respondents who chose "(4) Not at all" for all cases were classified as "Individuals without Psychological Distress"; on the contrary, respondents who chose choices (1) to (3) for any cases were classified as "Individuals with Psychological Distress".

### 4.2. Demographic Variables and Other Related Variables

Demographic variables of interest (dichotomized for analyses) included Gender (Male, Female), Race/Ethnicity (Non-Hispanic white, Racial and ethnic minority), Education (≤High school, >High school), Income Ranges (<\$20,000, ≥\$20,000), Geographic area (Non-metropolitan, Metropolitan), and Marital status (In marriage, Not in marriage), as well as Numerical demographic variables, including Age (continuous years) and BMI.

For further analysis, we selected as many variables as possible from the HINTS database that might be related to psychological distress by drawing on relevant literature and referring to historical experience. The potential independent variables we extracted were as follows: SeekCancerInfo (Yes, No), UseInternet (Yes, No), WearableDevTrackHealth (Yes, No), Social media user (Yes, No), RegularProvider (Yes, No), AccessOnlineRecord (Yes, No), Caregiving (Yes, No), OwnAbilityTakeCareHealth (Completely confident, Very confident, Somewhat confident, A little confident, Not confident at all), Deaf (Yes, No), TalkHealthFriends (Yes, No), MedConditions_Disease (Yes, No), Drink (Yes, No), Smoke (Yes, No), EverHadCancer (Yes, No), Cancercheck (Yes, No), FreqWorryCancer (Not at all, Slightly, Somewhat, Moderately, Extremely), and ModerateExerciseMinutes (Numerical demographic variables).

Table S1 presents the details of the above variables, including demographic variables and other independent variables, and information on reclassification.

## 5. Machine Learning Methods

### 5.1. Logistic Regression

Logistic regression was used to study the relationship between a dichotomous response variable, coded 0/1, and a set of explanatory variables $x_1, x_2, \ldots, x_n$ (categorical and numerical), which models the probability that $y$ belongs to a particular category [15].

$$p(y = 1 | x_1, x_2, \ldots, x_n) = \frac{e^{\alpha + \beta_1 x_1 + \ldots + \beta_n x_n}}{1 + e^{\alpha + \beta_1 x_1 + \ldots + \beta_n x_n}}$$

The odds ratio or likelihood ratio expresses the ratio between the probability $p$ that the dependent variable $y$ is 1 and the probability $1 - p$ that the dependent variable $y$ is 0. The natural logarithm of odds (Logit) is a linear function of the explanatory variables:

$$\log it(p) = \ln(\frac{p}{1 - p}) = \alpha + \beta_1 x_1 + \ldots + \beta_n x_n$$

In the above formula, $\beta_1, \beta_2, \ldots, \beta_n$ are the coefficients that measure the contribution of the independent variables $x_1, x_2, \ldots, x_n$ to $y$. If the coefficient $\beta$ is positive, $e^\beta > 1$ and the factor has a direct correlation with $y$; if $\beta$ is negative, $e^\beta$ is between 0 and 1.

### 5.2. Random Forests

The random forest is a multivariate statistical technique that considers an ensemble (forest) of trees for efficiency and predictive power [16]. Random forests use a bagging technique (bootstrap aggregation) to select a random sample of variables and observations at each tree node as the training dataset for model calibration. Since the random selection of the training dataset may affect the model's results, a large set of trees is applied to guarantee the stability of the model. Out-of-bag error is used to compute the model's error (OOBError) and establish the importance ranking of variables. This paper uses the "randomForest" function in the "randomForest" package to build a random forest for the psychological distress classification problem. The number of decision trees is set to 500, a fresh sample of two predictors was taken at each split, and the rest of the parameters are set to default values. The importance of the variable was judged according to the "Mean Decrease Gini" indicator, where the larger the value, the greater the importance of the variable.

### 5.3. ANN

Neural networks are algorithms that try to identify potential relationships in a dataset by mimicking the human brain function. Like the human brain structure, neural network models consist of neurons in complex and nonlinear forms. Neural networks have three basic types of layers: input layers, hidden layers, and output layers. Each neuron in the current layer is connected to the input signal of each neuron in the previous layer. In each connection process, the signal from the previous layer is multiplied by a weight, a bias is added, and then passed through a nonlinear activation function through multiple composites of simple nonlinear functions to achieve a complex input space to output space map. In this study, we used the neural network algorithm provided by the "nnet" R package. The input values are observations of 26 variables, and the output value is the probability of suffering from psychological distress. The "nnet" package sets a multinomial log–linear model, which is a feed-forward neural network with a single hidden layer. In addition, the ANN model is a feed-forward, five-fold cross-validated neural network containing automatically standardized variables. The five-fold cross-validation is to divide the data set into five subsets, with each subset used as a test set, while the rest are used as a training set. Cross-validation was repeated five times, and the average of the predicted values of the five times was used as the result. The model in this paper included a hidden layer with one node, with a decay parameter of 0.8. The purpose of the decay parameter is to prevent overfitting, so that the weights of each neuron converge to a small absolute

value. The variable importance of the model was measured by the combination of absolute values of coefficients.

*5.4. XGBoost*

Extreme gradient boosting is an efficient implementation of the gradient boosting framework from Chen and Guestrin (2016) [17]. In addition, XGBoost is a tree-based algorithm that belongs to supervised learning, which divides features based on the idea of a decision tree and limits the complexity of the tree. The input to the algorithm is also 26 variable observations to get the probability of suffering from psychological distress. We used this algorithm provided by the "xgboost" R package in this study. The package includes an efficient linear model solver and tree learning algorithms that can automatically perform parallel computing on a machine. It supports various objective functions, including regression, classification, and ranking. The package is quite flexible, so that the users are also allowed to define their own objectives easily. Furthermore, the models included 10-fold cross-validated algorithms containing automatically standardized variables. There are many parameters in the model. The number of boosting iterations means the number of decision trees. The learning rate can avoid overfitting and improve the robustness of the model by reducing the weight of the number. The parameters were set to the number of boosting iterations = 10, with a learning rate of 0.1, and other parameters were set to the default. The variable importance measurement method of this model was the same as a single tree (i.e., reduction in the loss function attributed to each variable at each split was summed over each node) but summed the importance estimates over each boosting iteration.

## 6. Results

The merged datasets from HINTS Cycle 3–Cycle 4 yielded a sample of 5484 respondents, including 2610 respondents without and 2874 respondents with psychological distress. Table 1 presents the frequencies and proportions of the variables. The Chi-squared test of categorical variables and the *t*-test of continuous variables showed significant differences in some variables between individuals with and without psychological distress ($p < 0.05$). Among the categorical variables, respondents choosing the following options comprised a significantly higher proportion ($p < 0.05$) in the group without psychological distress: "males," "had more than $20,000 annual income," "in marriage," "completely confident about self-health care," "ever had cancer," "never worry about getting cancer." For example, in this group, males accounted for 48.08% of the respondents, while in the group with psychological distress, the percentage decreased to 38.31%. The same was true for the other variables mentioned above: "had more than $20,000 annual income" (89.39% vs. 83.23%), "in marriage" (63.10% vs. 53.93%), "completely confident about self-health care" (31.61% vs. 16.95%), "ever had cancer" (7.62% vs. 6.26%), and "never worry about getting cancer" (24.79% vs. 12.91%). However, those choosing "ever looked for information about cancer" (50% vs. 58.28%), "using social media" (74.14% vs. 83.86%), "ever accessed online medical records" (44.14% vs. 46.49), "caring for or making healthcare decisions for someone" (14.06% vs. 18.20%), "self-health evaluation as good" (6.44% vs. 18.65%), "deaf" (5.36% vs. 7.34%), "had high blood pressure or other diseases" (53.68% vs. 69.31%), and "smoke" (35.71% vs. 44.64%) were significantly higher ($p < 0.05$) in the group with psychological distress. Among numeric variables, mean values of age and the average number of minutes of moderate daily exercise were significantly higher ($p < 0.05$) in the group without psychological distress, while BMI was significantly higher ($p < 0.05$) in the group with psychological distress. The Chi-squared test of categorical variables also showed no significant difference ($p > 0.05$) in some variables between individuals with and without psychological distress. In other words, the proportions of those variables were similar in the two groups. As for education, most individuals (approximately 81%) had above high school education in both groups.

The variables in the logistic regression were screened by three methods: forward selection, backward selection, and stepwise regression. The results obtained by the three variable selection methods were consistent. According to the variable *p*-value, Table 2 only retains the variables that were significant in the regression, and the crude odds ratios (ORs) and 95% CI are calculated to elucidate the effect of each variable on the psychological distress. Based on sociodemographic variables, relatively younger age (OR = 0.96, 95% CI: 0.96–0.97), unmarried (married OR = 0.65, 95% CI 0.54–0.78), non-Hispanic white (OR = 1.24, 95% CI: 1.03–1.48), and female (male OR = 0.68, 95% CI: 0.57–0.81) groups were more likely to have psychological distress. According to other research variables, those who searched for cancer-related information (OR = 1.34, 95% CI: 1.12–1.60), used social media (OR = 1.40, 95% CI: 1.11–1.77), were currently caring for or making health care decisions for someone with a medical, behavioral, disability, or other condition (OR = 1.34, 95% CI: 1.06–1.68), and believed they were in poor health (healthy OR = 0.55, 95% CI: 0.40–0.75) were more likely to experience psychological distress. Likewise, individuals who were more likely to experience psychological distress tended to be those who had hearing impairments (OR = 2.63, 95% CI: 1.82–3.79), had been told by a doctor or another health professional that they had health problems (OR = 2.27, 95% CI: 1.88–2.74), did not exercise (OR = 0.9978, 95% CI: 0.9961–0.9995), or smoked (OR = 1.33, 95% CI: 1.11–1.59). According to the multi-category variables (OwnAbilityTakeCareHealth and FreqWorryCancer), individuals who were less confident about taking care of their own bodies and more anxious about cancer were more likely to have psychological distress.

**Table 2.** Predictors of psychological distress identified using a logistic regression algorithm.

| Logistic Regression | |
| --- | --- |
| **Predictor** | **OR (95% CI)** |
| SeekCancerInfo (ref = No) | 1.34 (1.12, 1.60) |
| Social media user (ref = No) | 1.40 (1.11, 1.77) |
| Caregiving (ref = No) | 1.34 (1.06, 1.68) |
| GeneralHealth (ref = No) | 0.55 (0.40, 0.75) |
| OwnAbilityTakeCareHealth (ref = Completely confident) | |
|    Very confident | 1.46 (1.19, 1.80) |
|    Somewhat confident | 2.63 (2.02, 3.42) |
|    A little confident | 6.54 (3.24, 13.23) |
|    Not confident at all | 14.62 (3.03, 70.66) |
| Deaf (ref = No) | 2.63 (1.82, 3.79) |
| MedConditions_Disease (ref = No) | 2.27 (1.88, 2.74) |
| ModerateExerciseMinutes | 0.9978 (0.9961, 0.9995) |
| Smoke (ref = No) | 1.33 (1.11, 1.59) |
| FreqWorryCancer (ref = Not at all) | |
|    Slightly | 1.64 (1.27, 2.12) |
|    Somewhat | 2.02 (1.55, 2.61) |
|    Moderately | 2.09 (1.54, 2.85) |
|    Extremely | 2.19 (1.48, 3.24) |
| Age | 0.96 (0.96, 0.97) |
| Marital status (ref = not in marriage) | 0.65 (0.54, 0.78) |
| Race (ref = Racial and ethnic minority) | 1.24 (1.03, 1.48) |
| Gender (ref = Female) | 0.68 (0.57, 0.81) |

Table 2 shows significant predictors of psychological distress in the logistic regression, including SeekCancerInfo, Social media user, Caregiving, GeneralHealth, OwnAbilityTakeCareHealth, Deaf, MedConditions_Disease, ModerateExerciseMinutes, Smoke, FreqWorryCancer, Age, Marital status, Race, and Gender. According to different variable importance criteria, the random forests, ANN, and XGB can give the importance order of the relevant variables for predicting psychological distress. Table 3 lists the top 15 important predictors obtained under the random forests, ANN, and XGB, respectively. The three methods identified 20 different predictors, including 14 important predictors identified by

the logistic regression model, and another 6 predictors, namely, AccessOnlineRecord, Area, BMI, Drink, Income, and UseInternet.

**Table 3.** Predictors of psychological distress identified using Random Forests, ANN and XGB methods.

| Random Forests | | ANN | | XGB | |
|---|---|---|---|---|---|
| **Predictor** | **Importance** | **Predictor** | **Importance** | **Predictor** | **Importance** |
| Age | 100.94 | Age | 18.42 | Age | 100.00 |
| BMI | 80.77 | OwnAbilityTakeCareHealth | 12.94 | OwnAbilityTakeCareHealth | 99.44 |
| FreqWorryCancer | 62.90 | MedConditions_Disease | 12.29 | MedConditions_Disease | 42.85 |
| OwnAbilityTakeCareHealth | 62.25 | GeneralHealth | 6.24 | BMI | 30.13 |
| ModerateExerciseMinutes | 55.92 | FreqWorryCancer | 5.77 | FreqWorryCancer | 24.79 |
| MedConditions_Disease | 28.90 | Smoke | 5.37 | GeneralHealth | 15.78 |
| GeneralHealth | 23.34 | Deaf | 5.15 | ModerateExerciseMinutes | 11.76 |
| Gender | 20.89 | Gender | 4.93 | Smoke | 9.57 |
| Social media user | 20.73 | Caregiving | 3.29 | Gender | 7.75 |
| SeekCancerInfo | 19.85 | Social media user | 3.19 | Deaf | 5.74 |
| Marital status | 19.42 | SeekCancerInfo | 3.06 | SeekCancerInfo | 5.42 |
| Smoke | 18.11 | Income | 2.94 | Marital status | 4.15 |
| Drink | 17.20 | Marital status | 2.44 | Social media user | 3.30 |
| Race | 16.28 | Race | 2.39 | Caregiving | 2.67 |
| AccessOnlineRecord | 15.54 | UseInternet | 1.97 | Area | 2.48 |

The essence of this paper is a binary classification problem of judging whether an individual has psychological distress based on the set of inputs like SeekCancerInfo, Social media user, etc. Therefore, we evaluated the four machine learning methods covered in this article using a series of commonly used evaluation metrics for classification algorithms. Table 4 presents the accuracy, precision, recall, F1-score, and AUC values of the four machine learning methods on the validation set. Figure 1 shows the ROC curves and AUC values of four automated machine learning methods. Accuracy is a metric of a classification model that measures the number of correct classifications as a percentage of the total number of classifications made. Precision is the proportion of all positive classifications that are correctly classified, while recall is the proportion of total positive classifications that are correctly classified. The F1-score is the harmonic mean of precision and recall. According to the accuracy, recall, and F1-score values, the optimal model was random forests, with an accuracy of 67.83%, a recall of 72.70%, and an F1-score of 70.24%. However, from the perspective of the precision and AUC indicators, the optimal model was the ANN, with a precision of 70.02% and an AUC of 73.90%. AUC is not affected by the classification threshold and data distribution, and thus reflects the overall classification power of the model. Therefore, in general, this study preferred to choose the ANN as the optimal model to predict the risk of psychological distress.

**Table 4.** The correct classification metrics for each machine learning method.

| Criterion | Logistic Regression | Random Forests | ANN | XGB |
|---|---|---|---|---|
| Accuracy | 67.43% | 67.83% | 67.61% | 65.68% |
| Precision | 68.12% | 67.95% | 70.02% | 67.97% |
| Recall | 70.74% | 72.70% | 66.81% | 65.27% |
| F1-score | 69.41% | 70.24% | 68.38% | 66.59% |
| AUC | 73.54% | 73.15% | 73.90% | 70.84% |

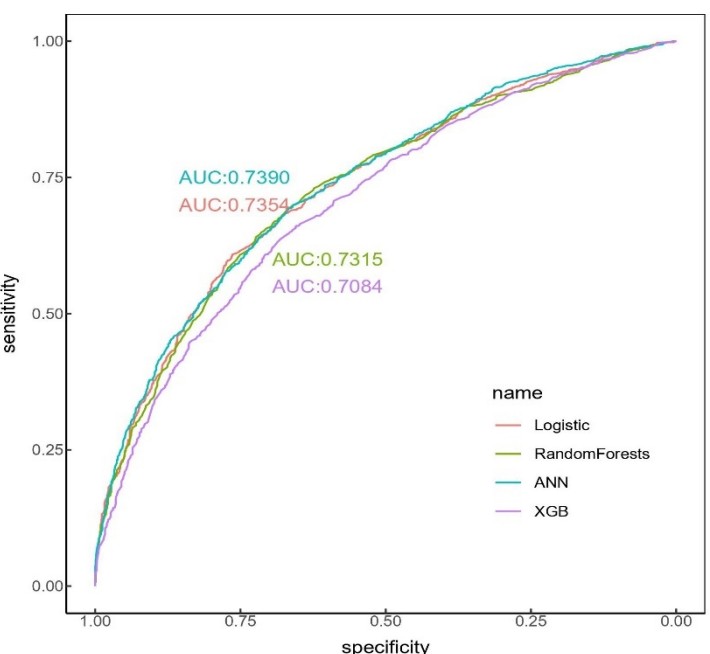

**Figure 1.** ROC curve of four machine learning methods.

## 7. Discussion

We conducted a comprehensive evaluation of the effects of individual sociodemographic characteristics, lifestyle, and behavioral habits on psychological distress. Although the generalization of the factors affecting individuals' psychological distress was difficult, based on the survey data, this study selected some variable sets with realistic and interpretable significance, demonstrating that an individual's psychological distress is related to their sociodemographic characteristics such as age and gender, lifestyle, behavioral habits, attention to health problems, etc.

Psychological distress is defined as the unpleasant feelings or emotions that a person may have when feeling overwhelmed. These emotions and feelings can interfere with daily routines and affect how the affected individual reacts to others. High levels of psychological distress indicate impaired mental health and may reflect common mental disorders, like depressive and anxiety disorders [18]. Psychological distress occurs when an individual faces stressors that they cannot cope with, including traumatic experiences, major life events, and everyday stressors such as workplace stress, family stress, interpersonal relationships, health issues, etc. Therefore, it is crucial to understand the factors contributing to psychological distress. This study provided ideas for predicting psychological distress based on personal behavior characteristics.

Self-report rating scales like the General Health Questionnaire [19] or MHI-5, derived from the RAND-36 questionnaire [20], are usually used to measure psychological distress levels. Based on the National Cancer Institute's 2019–2020 Health Information National Trends Survey (HINTS), we used the question "Over the past two weeks, how often have you been bothered by any of the following problems? (a) Little interest or pleasure in doing things, (b) Feeling down, depressed, or hopeless, (c) Feeling nervous, anxious, or on edge, and (d) Not being able to stop or control worrying" to determine whether a person suffers from psychological distress. Our analysis showed that approximately 52.41% of the population in the 5484 survey samples had symptoms of anxiety or depression.

Previous studies have mainly focused on sociodemographic differences in self-reported psychological distress or have divided individuals into different categories according to their characteristics to study some factors that affect their psychological distress. However, they have neglected the individual characteristics that generally affect psychological distress. To the best of our knowledge, this is the first study to apply a range of ma-

chine learning algorithms to a nationally representative sample to optimize psychological distress classification.

This study used four machine learning algorithms (logistic regression (linear), random forests (RF) (ensemble), the artificial neural network (ANN) (nonlinear), and gradient boosting (GB) (ensemble)) to identify and investigate factors affecting individuals' psychological distress. Twenty influencing variables concerning psychological distress were selected based on the coefficient significance in the logistic regression model and the variable importance indicators in the other three methods. Many well-established determinants were also identified as proof of concept for our analytical approach, such as sociodemographic characteristics [21]. While nonlinear and ensemble algorithms may exhibit better predictive performance than traditional parametric models, they are less interpretable [22]. Therefore, predictors determined by such algorithms should be evaluated in conjunction with relevant research evidence.

This study showed that sociodemographic indicators such as age, gender, education, marital status, race, area, and BMI significantly impacted psychological distress, while personal income did not significantly affect the prediction of psychological distress. In addition, predictors involving personal lifestyle and behavioral habits, such as smoking, drinking, exercise time, social network usage, etc., also play essential roles in predicting psychological distress. Finally, individuals' health status and their level of health concern were also associated with psychological distress. Generally speaking, people tend to be more prone to anxiety and psychological distress if they think they are unhealthy or have been told by a doctor that they have a medical condition.

The present research can provide a theoretical basis for screening individual mental health status and conducting mental health counseling. For example, the identified significant predictors can be used in psychiatric screening or electronic medical records, based on which machine learning algorithms can be applied to assess the likelihood of developing psychological distress. In this way, individuals who may have psychological distress can be identified in advance so to undergo mental health tests, thereby providing assistance to psychologists and other personnel.

There were some limitations in this study. Firstly, there were certain subjective factors in selecting candidate predictor sets related to psychological distress, and the relevant variables included might not have been comprehensive enough. Secondly, the relationship between the selected variables was not further studied, and there might be some collinearity in the screened important predictors. If used for linear regression analysis, there might be multicollinearity problems. Thirdly, the data were obtained using a self-report questionnaire. Therefore, we did not obtain detailed information on psychological distress, and self-report bias might have affected the results. Finally, the classification accuracy obtained by the machine learning method used in this paper should be further improved. In addition, the interpretability of the methods was poor. Further research is necessary to combine other methods to reveal the correlation or causal relationship between each predictor and psychological distress.

## 8. Conclusions

Based on the National Cancer Institute's 2019–2020 Health Information National Trends Survey (HINTS) database, this paper used four machine learning algorithms (logistic regression) (linear), random forests (RF) (ensemble), the artificial neural network (ANN) (nonlinear), and gradient boosting (GB) (ensemble) to identify and investigate the factors affecting psychological distress. These four models identified 20 variables as important predictors of psychological distress, consisting of 7 sociodemographic variables and 13 variables related to individual lifestyles and behavioral habits. As observed from the validation dataset fitting performance, our findings suggested that the optimal model was the ANN with an AUC value of 73.90%.

**Supplementary Materials:** The following supporting information can be downloaded at: https://www.mdpi.com/article/10.3390/pr10051030/s1, Table S1. Variables extracted from the HINTS database for this research.

**Author Contributions:** Methodology, L.L., J.L., L.Q., L.Y., J.Z., B.-C.S. and M.-C.C.; software, Y.C., X.Z. and Y.W.; formal analysis, Y.C. and X.Z.; writing—original draft preparation, Y.C., X.Z. and Y.W.; writing—review and editing, L.L., J.L., L.Q., L.Y., J.Z., B.-C.S. and M.-C.C. All authors have read and agreed to the published version of the manuscript.

**Funding:** This research was funded by Fu Jen Catholic University (A0110180 and A0110182) and University of International Business and Economics (UIBE) Huiyuan distinguished young scholars research fund (grant 20JQ07).

**Institutional Review Board Statement:** The study was conducted in accordance with the Declaration of Helsinki, and approved by the Westat Institutional Review Board of the U.S. National Institutes of Health Office of Human Subjects Research Protections.

**Informed Consent Statement:** All subjects participating in the study obtained informed consent.

**Data Availability Statement:** Publicly available datasets were analyzed in this study. This data can be found here: [https://hints.cancer.gov/data/download-data.aspx], accessed on 19 May 2022.

**Conflicts of Interest:** The authors declare no conflict of interest.

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
