# Peer review of "Machine Learning Methods to Identify Predictors of Psychological Distress"

_processes, doi:10.3390/pr10051030_

Round 1

Reviewer 1 Report

This manuscript has used four different machine learning algorithms for assessing psychological distress basing on the HINTs data with screened/recorded 26 variables. And it is claimed that 20 out of 26 variables had been confirmed by the study. The author mentioned that “Among the four machine learning models, ANN has the best predictive effect (AUC=73.90%).” Overall, the study is trying to “provide a theoretical basis for screening individual mental health status and

conducting mental health counseling”. But the novelty of the work is very limited, all those predicators are from HINTs data and have already been well studied and reported yet. Basing on what has been included in the manuscript, there is no new findings reported by this work. Although the authors had reduced the variables pool from 26 to 20, different variables were obtained when using different methods. And the results failed to confirm which machine learning algorithm is the most reliable one. For screening individual mental health status and conducting mental health counseling, the author has not mentioned how to weight those 20 variables in the assessment of psychological distress, either.

The major concerns about the manuscript are listed below:

  1. In page 6, row 110-113, “The variable “Psychological Distress” was calculated by the HINTS using the following four items. The first two items are for depression screening and the other two items for anxiety screening”. This paragraph is very confusing because it seems something is missing here. As the following sentences did not explain the theme in the beginning of paragraph.

  1. For the Demographic variables and other related variables introduce in Page 7 should be put together with the table 1 in page 3. It is hard to move back for checking the table when reading it in a different place. And it claimed that “We extracted more potential independent variables into the study”. What is the criterion for choosing these independent variables to study “Psychological Distress”? What are the reference for supporting your choices?

  1. In page 8, row 178-180, “The Chi-square test of categorical variables and the t-test of continuous variables showed a significant difference in some variables between people with and without psychological distress (P < 0.05).” Which P values is indicated here, p value in table 1? Table 1 has contained very basic and crucial information and it seems to be the only resource of the paper, but I have not seen any systematic introduction about it with all details throughout the paper. There is even no table notes for explaining P values shown in Table 1.

  1. What does Figure 1 mean in the paper? Which part of the manuscript does it belong to?

  1. In page 9, row 198-201, the statement “twenty predictors of psychological distress variables were identified by the 4 models as shown in Table 2 and Table 3.” Is not clear enough. Because there are different variables included in table 2 and table 3, and none of them has 20 variable included. Please stick to the results shown in the tables and introduce them one by one to make clear statement.

  1. For all those values shown in Table 4, all the values of each criterion are very similar to each other among  all four machine learning methods. Is there any significant difference  when using different machine learning methods? If there is, please mark clearly. If not, please offer the reason why it has been claimed in the abstract that “ANN has the best predictive effect” by only using AUC value. 

Reviewer 2 Report

In this paper, the authors used four machine learning algorithms to find predictors of psychological distress and timely diagnose the psychological distress.

In the introduction when you mention the previous studies, I recommend you to mention which are the traditional methods used in these studies. 

There is no reference for the  dataset used in this paper and it does not describe how the diagnosis of psychological distress was established.

The presentation of methods is blended with the results (Figure 1). 

The obtained results were not very accurate, so which methods did you use to improve them?

The limitations and a discussion about the ethical aspects of this study are not presented.

Reviewer 3 Report

This paper deals with the interesting topic of machine learning algorithms from the perspective of prediction of psychological distress. The main objective of the paper is to compare 4 different algorithms of machine learning ( logistic regression, random forest, neural networks and  gradient boosting) in term of their  effectiveness in identifying predictors of psychological distress.

Unfortunately the methodology and results presented in the paper are completely unclear. Authors very often use term “predictors” that in context of machine learning is associated with prediction problem. “Prediction” refers to the output of an algorithm after it has been trained on a historical dataset and applied to new data when forecasting the likelihood of a particular outcome. But this issue seems to be more of a classification problem. Moreover, the authors use the typical classification metrics like accuracy, recall, precision, F1-score and ROC curve for the evaluation. To rectify this problem, the following issues should be done:

1 – After Machine learning methods section authors should clearly explain methodology of research what means describe their algorithms (parameters, inputs and outputs, e.t.c.).

2 – As the research methodology is unclear, the results presented are also somewhat unclear - the authors should explain the metrics used and the results presented in Tables 2 and 3.

3 – Introduction is a bit skeptical and should be extended. The authors very briefly write about the use of machine learning methods in the applications of medical and psychological diagnostics - more references should be put in the paper. Moreover authors write that according to their best knowledge, this is the first study that applied a range of machine learning algorithms on so big representative sample to optimize psychological distress prediction. That is why authors should describe earlier approaches that were applied in this problem. It can be an extension of Introduction, or independent section e.g. Related works.

4 – In lines 155 – 163 the authors describe ANN and write about AlexNet architecture in context of the development of neural networks. But it is a specific type of neural networks - Convolutional Neural Network that is dedicated for computer vision and image processing, so I do not see associations with  MLP architecture that was presented in this paper.

5 – Some conclusions in section Discussion do not arise directly from the research results presented, but I believe that this problem will be solved by expanding the sections related to methodology and results.

Round 2

Reviewer 1 Report

Thanks for the clear point-to-point reply. It explains all my doubts on the result in the beginning. It can be seen clearly that the authors have carefully revised the manuscript according to my review comments. The revised manuscript have also corrected all the places I had addressed. It looks much much better after moving Fig1 with Table 4. Great revision.

Author Response

Thanks again for your previous comments and suggestions. These comments are of great help to the revision and improvement of this paper, and also have important guiding significance for our research. Also, thank you very much for your acknowledgment of our last revision. We are well aware that we have many problems in article design and description. Based on your guidance and suggestions, we hope to continue to improve our writing ability in the future, and to achieve more complete and detailed research. Thank you very much for your dedication and time.

Reviewer 2 Report

The authors carefully revised their manuscript and responded at my suggestions.

Author Response

(The authors gave the same response as above.)

Reviewer 3 Report

Thank you very much for the big amount of work that has been done by the authors to improve the quality of the article.

The extended version of the Introduction allows the reader to better understand the subject of the article and presents a wide spectrum of approaches that have been used in solving the problem of psychological distress prediction based on selected factors. In the last paragraph, the authors mention the database that was used in the conducted research. This is a good opportunity to highlight the authors' contribution, i.e. conducting the first research on this topic on such a large representative sample based on many different factors (stress predictors).

The machine learning algorithms used by the authors, as well as the metrics used to evaluate them, are also presented in much more detail. However, one sentence of the explanation still lacks in the text that the authors have already included in the cover letter sent in response to the comments. The explanation that it is a problem of binary classification based on the set of inputs like SeekCancerInfo, Social media user, e.t.c.

Author Response

 Thanks again for your previous comments and suggestions. These comments are of great help to the revision and improvement of this paper, and also have important guiding significance for our research.

In response to the question you raised in the "Introduction" section, we have added an explanation of the strengths of this study in the last paragraph of the Introduction. See row 85-87.

Furthermore, for the problem that "binary classification" has not been addressed in the text, we have made relevant supplements in the Abstract (See row 20-21)and explanation of Table 4 (See row 257-259).